Energy demands in high-intensity intermittent taekwondo specific exercises

http://orcid.org/0000-0002-9776-6015 Bartel Charles 1
http://orcid.org/0000-0001-5461-7119 Coswig Victor S. 2 3
http://orcid.org/0000-0003-4666-3531 Protzen Gabriel V. 1 4
http://orcid.org/0000-0003-3771-9660 Del Vecchio Fabricio B. 1 fabricioboscolo@gmail.com
1 Superior School of Physical Education, Universidade Federal de Pelotas , Pelotas, Rio Grande do Sul , Brazil
2 Physical Education and Sports Institute, Universidade Federal do Ceará , Fortaleza, Ceará , Brazil
3 Post Graduation Program in Human Movement Sciences, Universidade Federal do Pará , Castanhal, Pará , Brazil
4 Health Sciences Department, University of Santa Cruz do Sul, Santa Cruz do Sul , Rio Grande do Sul , Brazil
García-Ramos Amador
Electronic publication date: 2022 Aug 24
Publication date: 2022
Volume: 10
Electronic Location ID: e13654
Received 2021 Dec 6; Accepted 2022 Jun 9
Copyright: © 2022 Bartel et al.
Copyright year: 2022
Copyright holder: Bartel et al.
License: This is an open access article distributed under the terms of the Creative Commons Attribution License, which permits unrestricted use, distribution, reproduction and adaptation in any medium and for any purpose provided that it is properly attributed. For attribution, the original author(s), title, publication source (PeerJ) and either DOI or URL of the article must be cited.
License URL: https://creativecommons.org/licenses/by/4.0/

Keywords: Sport, Martial arts, Exercise physiology, Performance

Funding: The authors received no funding for this work.

==============================
Background

Taekwondo is an intermittent Olympic combat sport, which shows an aerobic predominance in matches and high participation of alactic metabolism for actions that determine competitive success. However, there is no information on energetic contribution systems in different high-intensity intermittent exercises for metabolic conditioning with specific movements. The study aimed to measure the physiological demands, mainly the energy expenditure, in taekwondo-specific high-intensity intermittent exercises (HIIE).

Methods

This study recruited ten male black belt athletes with a mean age of 20.2 ± 4 years, body mass of 62.8 ± 10.5 kg and height of 170.6 ± 7.8 cm, and total practice time of 11.8 ± 5.4 years. Subjects performed an incremental specific test and three different HIIE protocols on nonconsecutive days, and all comprised three 2-min rounds and 1 min of recovery between rounds. Heart rate, oxygen consumption, and blood lactate were measured. Energetic expenditure of aerobic, alactic, and lactic metabolisms was estimated through oxygen consumption, excess post-exercise oxygen consumption, and peak blood lactate after each round.

Results

For the mean of the three rounds, the TKDtest100 resulted in higher absolute and relative contribution from the aerobic metabolism (52.4 ± 4%; p = 0.01) and lower than the 35:5 relative alactic contribution (48.7 ± 5.4%; p = 0.03).

Conclusion

The mean of the three rounds for 35:5 and 15:10:5 presented similar absolute and relative contributions of aerobic and alactic metabolisms, whereas the TKDtest100 was a predominantly aerobic activity. We emphasize that aerobic metabolism was predominant from the second round in the 15:10:5 and 100%TKDtest protocols and in the last round of the 35:5 protocol.

Introduction

Taekwondo is an intermittent Olympic combat sport (Bridge et al., 2013), which shows an aerobic predominance in matches and high participation of alactic metabolism for actions that determine competitive success (Campos et al., 2012; Lopes-Silva et al., 2015). In combat simulations, the aerobic, anaerobic alactic, and anaerobic lactic metabolisms provide 66 ± 6%, 30 ± 6%, and 4 ± 2%, respectively, of the total energy required (Campos et al., 2012). As with other modalities, combat sports training includes a period destined for special physical preparation, which aims at practicing exercises in similar conditions to those of competition, considering their biomechanical and physiological specificity (Turner, 2011; Lyakh et al., 2014). Due to the difficulty of controlling training intensity in combat sports, protocols based on time-motion analysis are frequently used to mimic the demands found in combat (Vasconcelos et al., 2020).

Previously, similar hormonal, physiological, and physical responses were found between a specific kickboxing circuit training protocol and competition (Ouergui et al., 2015). On the contrary, in Taekwondo, several differences were found in metabolic, hormonal, cardiovascular, and neuromuscular responses when comparing simulated matches and a time-motion-specific training protocol (Bridge et al., 2013). However, both studies did not measure the contribution of energy systems and have no information concerning the participation of each energy source in these time-motion-based training protocols.

Despite these physiological differences, high-intensity interval training (HIIT) with general exercises has demonstrated beneficial adaptations in combat sports (Vasconcelos et al., 2020). Wrestlers who added two weekly sessions of repeated sprints to their 4-week training program showed an increase of 5.4% in maximal oxygen uptake (V̇O2MAX), 32.2% in time-to-exhaustion, and a significant increase in anaerobic power (Farzad et al., 2011). Similarly, an increase in aerobic power and anaerobic capacity were observed after 7 weeks of HIIT in karate (Ravier et al., 2009). However, both studies did not consider the motor gesture specificity while prescribing the exercises, considering that both used running in their performance. In Judo, after 4 weeks of comparing cyclic training vs specific gesture-based training protocols, HIIT promoted positive adaptations in the aerobic and anaerobic performance, regardless of the exercise model used (Franchini et al., 2016). Specifically in Taekwondo, similar adaptations in body composition (Ojeda-Aravena et al., 2021) and in some fitness components (Ojeda-Aravena et al., 2021; Ouergui et al., 2020) were evidenced while comparing general and gesture-specific HIIT protocols; although the specific protocol showed greater gains in agility and lower post training blood lactate concentration (Ouergui et al., 2020). This evidence reinforces that those physiological adaptations may appear similarly from both general and specific exercises. However, further studies are needed to confirm this evidence, especially after knowing the energetic contribution of different HIIT protocols.

Intermittent protocols with specific movements in combat sports aim to increase physical fitness through training methods using more similar motor and temporal actions than official matches (Franchini et al., 2016). In this sense, the effort to pause ratio (E:P) and the high- to the low-intensity ratio (Hi:Li) are important variables for developing physical conditioning exercises (Coswig, Ramos & Del Vecchio, 2016). In addition, it is indicated that specific protocols should be structured according to the time-motion structure found in real combat (Bridge et al., 2013; Ouergui et al., 2015) and applied intending to simulate physiological demands with no full contact with opponents (Crisafulli et al., 2009; Ouergui et al., 2015). This can be useful to minimize osteoarticular traumas through the use of padding, heavy bags, and other equipment, a strategy that could be considered for the days before the competitions (Coswig et al., 2016). This strategy could aid trainers in providing relevant physical and physiological stimuli while improving athletes’ technical and tactical strengths and weaknesses (Del Vecchio, Hirata & Franchini, 2011).

Understanding technical/tactical demands based on time-motion analysis and energy demands during a combat situation allows creating of replicable and straightforward high-intensity interval exercise (HIIE) protocols for physical training prescription (Del Vecchio, Hirata & Franchini, 2011; Ouergui et al., 2015). In addition, knowing the energetic and metabolic demands of the training methods is relevant to improve specificity in training prescription (Franchini, Panissa & Julio, 2013) and to provide relevant information on the load-recovery-adaptation spectrum (Banfi et al., 2012). Also, information on energetic contribution systems in different HIIE models for metabolic conditioning with specific movements for combat sports is quite scarce (Franchini et al., 2008; Lopes-Silva et al., 2015), and only three studies were conducted with combat simulation (Campos et al., 2012; Lopes-Silva et al., 2015, Julio et al., 2017). Therefore, the study aimed to characterize the physiological demands and the energetic system’s contribution to taekwondo-specific high-intensity intermittent exercises. The main hypothesis is that each protocol will induce different energetic contributions in the first round, while aerobic demands will became predominant along the session. If confirmed, it would allow coaches and trainers to prescribe HIIE models for specific energetic targets.

Materials and Methods

Participants

This experimental study, with repeated measures, recruited nine male participants, all black belts of the World Taekwondo (WT) International Federation, with a total practice time of 11.8 ± 5.4 years (Table 1). The following inclusion criteria were adopted: (i) non-smokers; (ii) not having any osteoarticular pain or lesions; (iii) not suffering from cardiovascular or metabolic diseases; (iv) to be graduated as Taekwondo black belt and have been regularly attending training sessions in the last 3 months at least three times a week, with daily sessions of 60 min or longer. Participants read and signed a free and informed consent form, and the research project was approved by the Federal University of Pelotas ethics committee (445.796/2013). An a priori sample size calculation was performed (G*Power 3.1) considering three metabolic systems and three repeated measures. Nine subjects would be needed to reach a statistical power of 0.84 considering effect size of 0.6.

Table 1 Sample characterization and performance in the specific incremental test.

	Mean ± SD	Minimum	Maximum	95% CI	
LL	UL	
Age (years)	20.2 ± 4	17	26	17.5	23.1	
Height (cm)	170.6 ± 7.8	157	180	165	176.1	
Body mass (kg)	62.8 ± 10.5	47.1	81.4	55.2	70.3	
VO2MAX (ml.kg−1.min−1)	58.5 ± 8.1	45.1	70.8	50.9	63.4	
VO2MAX (ml.min−1)	3,772 ± 450	3,084	4,550	3,455	4,066	
iVO2MAX (kicks/min)	34.4 ± 9	22	49	27.5	41.3	
HRMAX (bpm)	198 ± 4	189	203	194	200	
Note:

VO2MAX, maximal oxygen consumption; iVO2MAX, maximal oxygen consumption associated intensity; HRMAX, maximal heart rate; CI, confidence interval; LL, lower limit; UL, upper limit.

Experimental design and procedures

Firstly, the individuals filled out anamnesis, had their baseline heart rate and blood pressure parameters measured and had their weight (Filizola® scale, precision of 0.1 kg) and height (Sanny stadiometer, accuracy of 0.5 cm) registered. In the same session, they performed a taekwondo incremental test (TKDtest), and the order of training protocols was randomized by a simple draw.

The subjects performed the TKDtest, a previously validated procedure that showed mean differences of 2.2, 2.7, and 0.4 ml/kg/min for V̇O2PEAK, V̇O2 at the aerobic threshold, and V̇O2 at the anaerobic threshold, respectively, when compared to a treadmill test (Araujo et al., 2016). In the present study, the athletes performed semicircular kicks (Dolio-Tchagui) cued by a firing sound signal to a dummy’s torso (between the navel and nipple line) (BoomBoxe™, São Paulo, Brazil) at a height similar to the height of the evaluated participant. Briefly, the first stage has a frequency of ten kicks per minute, and three kicks are added at each stage. The test was interrupted when the individual reached exhaustion. The kicks were applied with the typical impact used by the athletes and were visually evaluated by a researcher. Athletes kept a self-selected average distance from the dummy, and the TKDtest was used to determine V̇O2MAX and its correspondent intensity (iV̇O2MAX ), expressed by kicks per minute.

Subsequent visits separated by a minimum of 48 h and a maximum of seven days were used to carry out the experimental protocols. Participants had an initial collection of a blood sample to measure lactate concentration ([La−]) and urea and performed a specific warm-up. Then, the participants rested for 6 min before the experimental protocol.

All training protocols were structured according to the current rules of the WT International Federation, constituting three 2-min rounds with 1 min of rest between them (3 × 2 min:1 min), which is the established time for official taekwondo matches. The subjects performed 7 min of warm-up, composed of 2 min of semicircular kicking, and stepping at 40% of the iV̇O2MAX (as identified on the first day after the TKDtest), 1 min at 50% of iV̇O2MAX, 1 min at 70% of iV̇O2MAX, and three sets of 30-s efforts at 100% of iV̇O2MAX with 30 s of passive recovery. A metronome controlled the pace relative to each range of relative intensity. The specific procedure for each high-intensity intermittent protocol is described below.

15:10:5 protocol

The 15:10:5 training protocol consisted of four effort blocks, lasting 15 s for observation (a period composed of rhythmic steps), 10 s for preparation (a period consisting of five semicircular kicks with a controlled frequency of one kick every 2 s), and 5 s of interaction (all-out semicircular kicks). The E:P of each round was 1:1 (15 s of observation:10 s of preparation plus 5 s of interaction).

100%TKDtest protocol

The 100%TKDtest training protocol was performed at 100% of iV̇O2MAX and was characterized by performing the kicking frequency identified in the TKDtest continuously throughout each 2 min round. The athletes performed rhythm steps between each semicircular kick applied and were requested to apply power as a match.

35:5 protocol

The 35:5 training protocol consisted of three 40 s blocks, where 35 s of observation and preparation periods were interspersed by 5 s of interaction, which consisted of all-out semicircular kicks. Thus, the E:P of each round was 1:7, a temporal structure similar to that previously described in Taekwondo matches (Campos et al., 2012), aiming to increase the specificity of the training protocol (Ouergui et al., 2015).

Blood samples

Capillary blood samples were drawn from puncturing the earlobe to measure [La−] and urea. [La−] was evaluated using 15 μl of blood and 30 μl of EDTA in Yellow Springs® 2300 Lactimeter (YSL, Cleveland, OH, USA) before each intervention, immediately after each round, and at minutes three, five, and seven after the completing the protocol. These samples were used to identify the kinetics and peak of [La−]. Urea was measured by analyzing 32 μl of capillary blood before and after the training sessions using a portable analyzer through reflectance photometry (Reflotron Analyser®; Boehringer-Mannheim, Lyon, France). All procedures followed the manufacturer’s recommendations.

Gas exchange data

An open-circuit gas analyzer (V̇O2000™; Medical Graphics, St. Paul, MN, USA) was used to estimate the average oxygen consumption every three breaths during the TKDtest and training efforts. Subjects used a silicone mouthpiece coupled to a high flow pneumotachograph during exercise and to a low flow pneumotachograph after the effort, with an umbilical attached to the equipment calibrated before and after the samples according to temperature status and auto-calibration process, as specified by the manufacturer. Telemetry acquired gas exchange data to provide greater comfort to the participants during their taekwondo movements.

Energetic contribution estimation

Energetic contribution is presented as absolute (kJ, kcal) and relative (%). It refers to how much energy was spent in each protocol and its respective energy source (Protzen et al., 2020). V̇O2 was continuously measured with the average of three ventilation during the efforts in the training protocols and was used to estimate aerobic metabolism energy (WAER) (Bertuzzi et al., 2007). These values were expressed over rest values obtained 5 min before the beginning of the protocol when the subjects remained in an orthostatic position.

To estimate the contribution of the anaerobic lactic system for energy production (W[La−]), it was assumed that the accumulation of 1 mmol.L−1 is equivalent to 3 ml.O2.kg−1 of body mass (Campos et al., 2012).

EPOCFAST was used to estimate the contribution of the anaerobic alactic system (WPCR). Previous studies have applied such procedures (Bertuzzi et al., 2007; Campos et al., 2012) and followed previously described assumptions (Di Prampero & Ferreti, 1999). Free online software (GEDAE-Lab) was used to calculate contributions. The procedure was tested and validated against standard calculations, with an intraclass correlation coefficient of 0.95 for energy expenditure and energy contribution calculation (Bertuzzi et al., 2016). The delta [La−] was calculated based on the variation between rounds.

Statistical analysis

Data are presented as mean ± standard deviation after checking the parametric distribution. A two-way analysis of variance for repeated measures (round × protocol) was used. The Mauchly test was used to check data sphericity, and the Greenhouse-Geisser correction was applied when necessary. Significant differences between rounds were located with Bonferroni posthoc, while differences between training protocols and metabolisms were located using Scheffé posthoc. The IBM SPSS software (22.0; Chicago, IL, USA) was used, and a significance level of p < 0.05 was adopted.

Results

Table 2 indicates that there were statistically significant differences in the physiological responses between training protocols (F = 3.126; p < 0.001, η2p = 0.44) and between metabolisms in the same protocol (F = 3.791; p < 0.001, η2p = 0.245).

Table 2 Physiological responses during each round of the three training protocols. Data shown as mean and (standard deviation).

	15:10:5	TKDtest100	35:5	
	R1	R2	R3	R1	R2	R3	R1	R2	R3	
WAER	
kJ	59 (12)ǁ‡	92 (13)*#‡	101 (20)*ǁ‡	70 (11)‡	105 (17)*ǁ‡	110 (16)*ǁ‡	48 (13)#ǁ‡	95 (17)*‡	97 (17)*#‡	
Kcal	14 (3)ǁ‡	22 (3)*#‡	24 (4)*ǁ‡	17 (3)‡	25 (4)*ǁ‡	26 (4)*ǁ‡	12 (3)#ǁ‡	23 (4)*‡	23 (4)*#‡	
%	37.6 (2.1)#ǁ‡	52.9 (6.8)*ǁ‡	56.5 (6.9)*ǁ‡	43.8 (5.1)‡	55 (6)*ǁ‡	58.5 (2.7)*ǁ‡	34.5 (7.8)#ǁ‡	50.8 (5.9)*‡	53.8 (5.4)*ǁ‡	
WPCR	
kJ	79 (28)‡	77 (29)‡	75 (28)‡	79 (16)‡	80 (14)‡	74 (22)‡	87 (27)‡	85 (32)‡	80 (19)‡	
Kcal	19 (7)‡	18 (7)‡	18 (7)‡	19 (4)‡	19 (3)‡	18 (5)‡	21 (6)‡	20 (8)‡	19 (5)‡	
%	49.7 (5.1)@‡	42.6 (7.1)‡	41.1 (7.1)*‡	49.1 (5.6)@‡	41.7 (3.1)‡	38.6 (5.9)*‡	57.9 (8)‡	44.1 (7.2)*‡	43.9 (4.1)*‡	
W[La−]	
kJ	20 (6)	8 (5)*	4 (3)*	11 (3)§	7 (3)	6 (3)	11 (4)§	10 (6)	4 (6)	
Kcal	5 (1)	2 (1)*	1 (1)*	3 (1)§	2 (1)	1 (1)	3 (1)§	2 (1)	1 (1)	
%	12.7 (3.6)	4.5 (2.1)*	2.4 (1.4)*	7.1 (2.1)§	3.3 (1.2)	2.9 (1)*	7.6 (2.4)§	5 (2.8)	2.3 (2.7)*	
[La−]PEAK (mmol.L−1)	5.8 (1.7)	7.6 (1.6)	8.3 (0.9)*	3.8 (1)§	5.4 (1.2)§	6.7 (1.2)*§	3.6 (1.2)§	6 (2.3)*§	6.7 (2.4)*§	
∆[La−] (mmol.L−1)	4.8 (1.5)	1.8 (1.3)*	0.7 (0.9)*	2.8 (1)§	1.5 (0.6)	1.2 (0.5)	2.6 (1.1)	2.4 (1.4)†§	0.4 (0.3)†	
VO2PEAK (mL.min−1)	3163 (606)@	3157 (354)	3205 (417)	3226 (501)@	3335 (614)	3363 (582)	2763 (448)	3058 (467)	3149 (439)	
%VO2MAX	83.5 (10.1)@	83.9 (6.0)	85.1 (7.3)	85.6 (9.7)@	88 (9)	89 (8.5)	73.3 (7.3)	81.4 (10.2)	83.8 (9.6)	
Notes:

WAER, aerobic energy; WPCR, alactic energy; W[La−], lactic energy; kJ, kilojoule; Kcal, kilocalorie; [La−]PEAK, peak blood lactate concentration; ∆[La−], delta of variation of blood lactate; VO2PEAK, peak oxygen consumption; %VO2MAX, percentage of maximal oxygen consumption.

* Different from round 1 for the same training.

† Different from round 3 for the same training.

§ Different from protocol 15:10:5 for the same round.

# Different from the TKDtest100 protocol for the same round.

@ Different from the 35:5 protocol for the same round.

‖ Different from alactic metabolism.

‡ Different from lactic metabolism.

In the first round, the 15:10:5 training protocol required greater lactic metabolism activity (absolute and relative) compared to 100%TKDtest (p < 0.001) and the 35:5 (p < 0.001). 100%TKDtest provided a greater relative energy contribution from aerobic metabolism than 15:10:5 (p = 0.01) and 35:5 (p < 0.001) protocols for the same moments. Moreover, it required higher absolute aerobic energy production than round 2 of the 15:10:5 training protocol (p = 0.02) and rounds 1 and 3 when compared to the 35:5 protocol (p < 0.001 and p = 0.02, respectively). The 35:5 provided higher relative energy intakes of alactic metabolism compared to the 15:10:5 and 100%TKDtest protocols during round 1 (p = 0.002 and p = 0.001, respectively).

For the mean of the three rounds, the 100%TKDtest obtained higher absolute and relative contribution of the aerobic metabolism (p = 0.01) and lower relative alactic contribution (p = 0.03) than the 35:5 (Figs. 1 and 2). A lower relative and absolute contribution of lactic metabolism were found in all rounds of all training protocols related to aerobic and alactic (p < 0.001 for all). Alactic contribution was higher during the first round in the 15:10:5 and 35:5 protocols when compared to the aerobic contribution (p < 0.001), with no statistical difference in 100%TKDtest (p = 0.07). In the second and third rounds, higher aerobic than alactic contribution was demonstrated in the 15:10:5 protocols and 100%TKDtest (p = 0.001 and p < 0.001, respectively). However, in the 35:5 protocol, there was no difference between the two energy systems in the second round (p = 0.8), only in the third round (p = 0.04).

Figure 1 Average contribution of bioenergetic systems of all three protocols.

# = different from TKDtest100 for the same energy system ǁ = different from WPCR ‡ = different from W[La−].

Figure 2 Total energy expenditure of all three protocols.

# = different from TKDtest100 for the same energy system ǁ = different from WPCR ‡ = different from W[La−].

Differences were found between training protocols for V̇O2PEAK (F = 2.86; p = 0.04; η2p = 0.07) and V̇O2MAX% reached (F= 5.74; p = 0.005; η2p = 0.14). It is observed that during the first round, the 15:10:5 and 100%TKDtest protocols induced greater increases in V̇O2 when compared to 35:5 (p = 0.01 and p = 0.003, respectively).

[La−] was different between training protocols (F = 12.35, p < 0.001, η2p = 0.25). Posthoc analysis shows that it was higher at all moments during the 15:10:5 (p < 0.05). It can also be highlighted that the [La−] peak was higher than the values observed after the first round in all protocols (p < 0.01). Finally, the 15:10:5 protocol delta (which induced higher [La−]) was statistically superior to the other protocols during round 1 (p < 0.001) (Table 2). No differences in urea values were observed between moments and protocols. Pre- and post-exercise values were 27 ± 4.6 mg.dL−1 and 28.3 ± 4.3 mg.dL−1 for 15:10:5 protocol; 26.4 ± 5.1 mg.dL−1 and 27.4 ± 5.8 mg.dL−1 for TKDtest100 protocol; and 26.8 ± 4.9 mg.dL−1 and 24.3 ± 3.8 mg.dL−1 for 35:5 protocol, respectively.

Discussion

The present study aimed to characterize and compare the physiological demands of three HIIE protocols employing specific taekwondo movements. First, we note that in the average of the three rounds, the 100%TKDtest protocol provided a predominance of aerobic metabolism. At the same time, no differences were observed for aerobic and alactic contributions in 15:10:5 and 35:5. The [La−] reached higher values in the 15:10:5 protocol, while the 35:5 induced lower V̇O2 values than the other protocols. To the authors’ knowledge, this is the first study evaluating the request of the energy systems using different HIIE protocols in combat sports; our findings indicate that, despite showing different temporal relationships, the proposed protocols induced physiological stress in a similar magnitude to that found in taekwondo combat simulations (Campos et al., 2012); however, with a distinct contribution from energy systems.

Higher glycolytic participation (absolute and relative) in the 15:10:5 protocol during the first round may have occurred due to the intermittent structure of the efforts when compared to the 100%TKDtest and the longer duration of the exercise blocks when compared to the 35:5 (Balsom et al., 1992a; Bussweiler & Hartmann, 2012). This finding is supported by a previous study that analyzed the [La−] production in repeated sprints distributed in different ways: (i) 40 × 15 m (S15), 20 × 30 m (S30), and 15 × 40 m (S40), with identical distances traveled (~600 m) and intervals (~30 s) between each set. After the end of the exercise, the mean [La−] observed were 6.8 ± 1.5 mmol.L−1, 13.9 ± 1.7 mmol.L−1, and 16.8 ± 1.1 mmol.L−1 for S15, S30, and S40, respectively, indicating that longer efforts in HIIT may induce greater glycolytic activation (Balsom et al., 1992a).

The higher (absolute and relative) contribution from aerobic metabolism in the 100%TKDtest protocol in comparison to the 35:5 protocol can also be explained by the prolonged duration of the efforts (2 min vs 15 s, per round) since aerobic metabolism is fundamental for supplying energy in longer duration activities (Balsom et al., 1992b; Gastin, 2001). The higher relative alactic contribution of the 35:5 protocol during the first round when compared to 15:10:5 and 100%TKDtest and the total mean over 100%TKDtest can be explained by the lower number of high-intensity effort stimuli and high recovery time imposed by 35:5, which contributes to preserving and partially resynthesizing high energy phosphates (Balsom et al., 1992a; Gaitanos et al., 1993), considering that a higher rate of depletion occurs only in longer high-intensity effort blocks (Campos et al., 2012). This is also elucidated by previous studies indicating that manipulating variables such as the duration of effort time (Balsom et al., 1992a; Bussweiler & Hartmann, 2012) and the stimuli amount (Gaitanos et al., 1993) is related to a predominance of distinct energy systems, and generally, as these variables increase, the demand for aerobic metabolism also increases (Glaister, 2005).

When analyzing the energy systems, it is suggested that the 15:10:5 protocol has a higher lactic characteristic compared to the 100%TKDtest and 35:5, even though this energy system was not predominant. In addition, we emphasize that there was an increase in the absolute and relative aerobic contribution from the second round, which determines its predominance. This is associated with the fact that the aerobic contribution continuously increases throughout the high-intensity exercise, whether it is continuous (Gastin, 2001) or intermittent (Glaister, 2005). There is a simultaneous decrease in the activity of the anaerobic component (Gaitanos et al., 1993), which can be explained due to the inability to complete high energy phosphate restoration during a 1-min recovery interval (Campos et al., 2012) and also by aerobic metabolism activation to restore homeostasis (Glaister, 2005) and resynthesize phosphocreatine (PCr) (McMahon & Jenkins, 2002) during the low-intensity actions.

Previous studies on taekwondo fights show a large variability of [La−] (from 2.9 ± 2.1 to 11.9 ± 2.1 mmol.L−1), which indicates a high variation in glycolytic metabolism contribution (Butios & Tasika, 2007; Bridge, Jones & Drust, 2009). [La−] was higher in the 15:10:5 protocol at all times of the combat, but there was only a difference in Δ [La−] at the end of the first round, indicating that glycolytic activation was only greater in the first round. Moreover, we observed that [La−] was superior in round 3 compared to round 1 in all protocols. However, the increase of this metabolite throughout the rounds does not indicate increased lactic contribution, considering that ∆ [La−] points suggest a decrease in lactate production and consequent reduced glycolytic participation throughout HIIE efforts (Campos et al., 2012). The [La−] of 100%TKDtest and 35:5 protocols were similar to the values reported by Campos et al. (2012) and Lopes-Silva et al. (2015), indicating similarity in lactic metabolism activation. All protocols induced higher [La−] than that observed in a previous study that reported 3.6 ± 2.7 mmol.L−1 after performing a specific exercise with similar duration to the present study (Bridge et al., 2013). This was probably because the average duration of the high-intensity efforts was relatively short (~1.3 s) (Balsom et al., 1992a). Therefore, it is possible to infer that the protocols tested in our investigation present greater specificity from the point of view of glycolytic metabolism.

In this study, we have also sought to understand the acute impact of HIIE with specific movements on urea concentration, which is considered a protein catabolism marker (Viru, 1994). Urea values may be relevant during the training process since excessive protein degradation is undesirable, especially in the competitive phase (Viru & Viru, 2001). However, it is noteworthy that no training protocol generated an increase in urea concentrations, indicating that executing the protocol only once does not induce prominent protein catabolism (Viru, 1994). The concentrations found in the present study are substantially different from the value reported in a previous study with Muay-Thai fighters (29.6 ± 3.1 mg.dL−1); nonetheless, athletes were undergoing intensive training, which may increase protein catabolism (Saengsirisuwan, Phadungkij & Pholpramool, 1998). New studies should consider analyzing different physiological markers after successive repetitions of the training protocols suggested in the present study, for two main reasons; the athletes compete in three to five matches to win medals in national and international championships, and successive repetitions of physical stimuli in daily training were performed with three to six repetitions per training session.

Some limitations should be considered while interpreting our findings. First, subjective effort perception was not measured, which could highlight differences between protocols when it is not evidenced in metabolic and mechanical parameters. Second, measures before the execution of each exercise would be of interest to infer recovery status and physical readiness; however, we believe that the randomization of exercises order may have reduced the impact of this issue.

Regarding practical applications, our findings demonstrated that it is possible to promote physiological stress, in a comparable level of taekwondo matches, with specific HIIE. It means that this kind of protocol can be considered an option for physical fitness development without the need to use general and non-specific exercises with traditional training methods. Moreover, when analyzing the contributions of each round, it is observed that manipulations in the exercise variables (mainly effort and pause duration and intensity) can be made to stimulate greater demand for an energy source. Thus, the 100%TKDtest protocol is indicated when the session aims to raise the aerobic component (power and capacity). In contrast, the 35:5 protocol could, for example, include 3–5 mins of recovery intervals between rounds, thus promoting greater ATP-CP system resynthesis and anaerobic power stimuli. The use of 15:10:5 may be effective in the maintenance period since aerobic and alactic metabolism activation are similar. Therefore, it can be inferred that each protocol can be used for developing or maintaining specific metabolic components according to the training objective.

Conclusions

Among the specific HIIE for Taekwondo proposed in the present study, we can conclude that the mean of the three rounds for 35:5 and 15:10:5 presented similar absolute and relative contributions of aerobic and alactic metabolisms. In contrast, the 100%TKDtest was a predominantly aerobic activity. Furthermore, for the first round, the 100%TKDtest protocol induced greater aerobic contribution, 35:5 demanded greater alactic contribution, while the 15:10:5 protocol promoted greater lactic system activation compared to the other protocols. We emphasize that aerobic metabolism was predominant from the second round in the 15:10:5 and 100%TKDtest protocols, and in the last round of the 35:5 protocol.

Supplemental Information

Supplemental Information 1 Raw data spreadsheet containing the presented and additional analysis.

Individual and group data of energetic contributions after different HIIT taekwondo protocols

Click here for additional data file.

We thank the Ph.D. Marlos Rodrigues Domingues and Ph.D. Airton José Rombaldi for the critical review of the manuscript.

Additional Information and Declarations

Competing Interests

Author Contributions

Human Ethics

Data Availability

The authors declare that they have no competing interests.

Charles Bartel conceived and designed the experiments, performed the experiments, analyzed the data, prepared figures and/or tables, authored or reviewed drafts of the article, and approved the final draft.

Victor S. Coswig analyzed the data, prepared figures and/or tables, authored or reviewed drafts of the article, and approved the final draft.

Gabriel V. Protzen performed the experiments, analyzed the data, authored or reviewed drafts of the article, and approved the final draft.

Fabricio B. Del Vecchio conceived and designed the experiments, performed the experiments, analyzed the data, prepared figures and/or tables, authored or reviewed drafts of the article, and approved the final draft.

The following information was supplied relating to ethical approvals (i.e., approving body and any reference numbers):

The research project was approved by the Federal University of Pelotas ethics committee (445.796/2013)

The following information was supplied regarding data availability:

The raw measurements are available in the Supplemental File.

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
