# Peer review of "Energy demands in high-intensity intermittent taekwondo specific exercises"

_PeerJ, doi:10.7717/peerj.13654_

## Round 0.1 · original submission · Major Revisions

Dear authors, reviewers provided good comments regarding your paper. However, there are several aspects of the paper that should be modified or clarified following the reviewer´s comment prior to the acceptance of the paper.


·

Basic reporting

The article include sufficient introduction and background to demonstrate how the work fits into the broader field of knowledge. Relevant prior literature was appropriately referenced. This is a well-written paper.

Experimental design

Methods were described with sufficient information to be reproducible by another investigator. The investigation was conducted rigorously and to a high technical standard. The research was conducted in conformity with the prevailing ethical standards in the field. However, exactly how the subject group was determined has not been explained. This section can be elaborated on.

Validity of the findings

Conclusions are well stated, linked to the original research question and limited to supporting results.

Additional comments

Congratulations on your work. This is a well-written paper.

·

Basic reporting

The present article includes sufficient background with relevant literature references. An English revision is required for the paper. The article is well structured and falls with the standards format of the journal, represents all the results, that are clearly presented.

Experimental design

- The authors reported that the use of high-intensity training with TKD specific movements and generic exercises induced similar adaptations. However, they study the energy contribution of HIIE; it is a question of rationale! Could the authors justify more their rationale and what they will bring as new compared to other studies, what is new in term of exercises’ structure and how this will help in training?
-L74: not all fitness components (for example agility was higher in specific training group compared to generic one in two tests) and that blood lactate post training was lower in specific group compared to generic one? Please change accordingly.
L76: reinforce
- Please add hypotheses at the end of the introduction section.
Materials & Methods
Participants
-It is only called World taekwondo (WT) is not?
- The authors should report a sample size calculation as the sample is relatively small.
- What do the authors mean by “having a black belt in the modality?”
Experimental design and procedures
- How the randomization was performed, a brief description can be added.
How the authors knew that athletes performed warm-up at the intensity desired while it is determined in kicks/min, I would ask to have an idea about how to organize this and how athletes paced their kicks execution to reach the intensity desired?
- Do the authors clearly describe based on what they choose to use these specific exercises, E/P ratios, specific techniques, why they used the 100%TKDtest Protocol? As I am not really impressionned with these exercise as while there are two exercises that are similar and differ in term of their time-structure, the other exercise (100%TKDtest Protocol) is really different from the two others?
- - Why the authors did not use some subjective effort measures like RPE , which may highlight differences between protocols when physiological and metabolic parameters fail to find it ?

Validity of the findings

The study presents such novelty to the literature of combat sports, with all results presented, from data that was well analyzed with appropriate statistical methods. The conclusions support partially the results found and should be improved.

Additional comments

General comments
The authors investigated the metabolic demands of specific TKD high-intensity exercises. The study can bring additional information to direct specifically training. Authors are encouraged to conducted changes and some issues should be addressed before acceptance.
Specific comments:
Abstract:
L22: please clarify which specific actions can be supported by alactic metabolism.
L23: however, there is no information available on energy contribution ….
L29: could the authors describe briefly what are the three exercises performed?
Results:
L35: TKDtest100? This may confirm my suggestion to explain exercises and provide specific abbreviations.
- I think this part should be improved more, as for me I did not understand this energy contribution regarding different exercises.
Conclusion:
The conclusion supports some findings that are not presented in the results’ section. Please change accordingly to ensure harmony between parts.

Introduction:
The introduction is well written with good choice of supporting literature.
- The authors reported that the use of high-intensity training with TKD specific movements and generic exercises induced similar adaptations. However, they study the energy contribution of HIIE; it is a question of rationale! Could the authors justify more their rationale and what they will bring as new compared to other studies, what is new in term of exercises’ structure and how this will help in training?
-L74: not all fitness components (for example agility was higher in specific training group compared to generic one in two tests) and that blood lactate post training was lower in specific group compared to generic one? Please change accordingly.
L76: reinforce
- Please add hypotheses at the end of the introduction section.
Materials & Methods
Participants
-It is only called World taekwondo (WT) is not?
- The authors should report a sample size calculation as the sample is relatively small.
- What do the authors mean by “having a black belt in the modality?”
Experimental design and procedures
- How the randomization was performed, a brief description can be added.
How the authors knew that athletes performed warm-up at the intensity desired while it is determined in kicks/min, I would ask to have an idea about how to organize this and how athletes paced their kicks execution to reach the intensity desired?
- Do the authors clearly describe based on what they choose to use these specific exercises, E/P ratios, specific techniques, why they used the 100%TKDtest Protocol? As I am not really impressionned with these exercise as while there are two exercises that are similar and differ in term of their time-structure, the other exercise (100%TKDtest Protocol) is really different from the two others?
- - Why the authors did not use some subjective effort measures like RPE , which may highlight differences between protocols when physiological and metabolic parameters fail to find it ?
Statistical analysis
- Please add the statistical software used.
- Reporting effect size and confidence intervals are highly recommended.
- Do the authors verify that baseline measurement were the same for athletes before realizing exercises?
Discussion
- The authors discussed well the results.
- Please add limitations at the end of the discussion.
- Please add some practical applications and how the results of the study are useful for coaches and what this study differently brings from existing studies.

---

## Round 0.2 · accepted · Accept

The reviewers´comments were properly addressed.